# Rating enrichment items by female group-housed laboratory mice in multiple binary choice tests using an RFID-based tracking system

Ute Hobbiesiefken[1], Birk Urmersbach[1], Anne Jaap[1], Kai Diederich[1], Lars Lewejohann[1,2]*

1 German Center for the Protection of Laboratory Animals (Bf3R), German Federal Institute for Risk Assessment (BfR), Berlin, Germany, 2 Animal Behavior and Laboratory Animal Science, Institute of Animal Welfare, Freie Universität Berlin, Berlin, Germany

* Lars.Lewejohann@bfr.bund.de

**Data Availability Statement:** The authors confirm that the data supporting the findings of this study are available within the article and its Supporting

## Abstract

Laboratory mice spend most of their lives in cages, not experiments, so improving housing conditions is a first-choice approach to improving their welfare. Despite the increasing popularity of enrichment, little is known about the benefits from an animal perspective. For a detailed analysis, we categorized enrichment items according to their prospective use into the categories 'structural', 'housing', and 'foraging'. In homecage-based multiple binary choice tests 12 female C57BL/6J mice chose between enrichment items within the respective categories over a 46-hour period. A new analyzing method combined the binary decisions and ranked the enrichment items within each category by calculating worth values and consensus errors. Although there was no unequivocal ranking that was true in its entire rank order for all individual mice, certain elements (e.g. lattice ball, second plane) were always among the top positions. Overall, a high consensus error in ranking positions reflects strong individual differences in preferences which could not be resolved due to the relatively small sample size. However, individual differences in the preference for enrichment items highlights the importance of a varied enrichment approach, as there does not seem to be one item that satisfies the wants and needs of all individuals to the same degree. An enrichment concept, in which the needs of the animals are central, contributes to a more specific refinement of housing conditions.

## Introduction

Attitudes toward animals as fellow living creatures have changed significantly in recent decades. There is growing concern about the conditions under which laboratory animals are kept, and it is therefore not surprising that legal requirements are also becoming increasingly demanding. In Europe, minimum requirements for housing laboratory animals are set out in Directive 2010/63/EU [1], which stipulates that animals must be housed according to the

Information materials. 3D printing templates of self-designed enrichments used in this study are openly available in Github under: https://github.com/RefinementReferenceCenter/MoPSS-preference-test-supplement-new.

**Funding:** The funders had no role in study design, data collection and analysis, decision to publish, or preparation of the manuscript.

**Competing interests:** The authors have declared that no competing interests exist.

specific needs and characteristics of each species. Experimental animals should be provided with 'space of sufficient complexity to allow expression of a wide range of normal behavior' [1]. While the available space itself is a pressing issue for future improvements, the issue of complexity is usually approached through what is known as 'enrichment of housing conditions'. It is reasonable to assume that additional enrichment opportunities in barren cages will create a more complex environment, which is likely to be appreciated by the animals [2,3] and they are even willing to work for access to enrichment opportunities [4].

However, it is important to note that 'enrichment' has become an umbrella term that encompasses a wide variety of different elements. Therefore, it must be kept in mind that by no means a uniformly accepted enrichment is meant when speaking of effects of enrichment [2,3,5]. This being said, many research groups have indeed shown the benefits of enriched environments relative to conventional housing on well-being parameters in mice [3,6]. Abnormal repetitive behavior expression, behavioral measures of anxiety, as well as growth and stress physiology were influenced positively by providing mice with a more varying environment using enrichment items [7]. Access to enrichment leads to improved learning and memory function [8,9], increased hippocampal neurogenesis [9,10], attenuated stress responses and enhanced natural killer cell activity [11]. Importantly, studies showed no generalizable influence of a more diverse environment on variability of important parameters in biomedical research in mice [7,12,13]. With regard to the workload of animal caretakers only a slight increase was noted while their overall assessment of providing enrichment in light of enhanced well-being for laboratory rodents was reported as good [14]. Overall, there is increasing evidence that keeping animals in conventional housing conditions may be a negative factor in the development of behavioral disorders because of its impoverished character [15]. Thus, it must be borne in mind that mice from suboptimal housing conditions which show abnormal behavioral and brain development render a less predictive animal model for biomedical research.

To create a more varied and stimulating environment, the size of the home cage can be enlarged, the group size increased, and stimulating elements can be provided [16,17]. However, the human perspective does not necessarily reflect the wants and needs of mice [2]. Therefore, it is essential to ask the animals themselves about the adequacy of the enrichment items [18,19]. To determine how different items are perceived by the animals themselves [20], animal centric strategies like preference tests will help to assess and rate different items [19,21–23].

From the three typically used preference testing designs [22], T-Maze, conditioned place preference, and home cage based preference tests, the last one seems to be the most appropriate for rating enrichment items. Especially when it comes to the avoidance of frequent animal handling and the opportunity to extend testing periods up to a full circadian cycle or longer [22]. Additionally, choice tests conducted within the home cage without the influence of an experimenter correspond better to real laboratory keeping conditions [24,25]. Home cage based testing systems usually consist of two [26,27] or more [26,28] connected cages with or without a center cage. In such tests mice are able to stay in their preferred surroundings and the cage that is chosen with the longer period of stay is regarded as the preferred one, or, in case of aversive properties, as the one least avoided [22].

For our preference test, we used the Mouse Position Surveillance System (MoPSS), a new test system designed and constructed in our laboratory [29] to ask for enrichment item preferences in female C57BL/6J mice, a widely used strain in biomedical research [30]. The MoPSS allows automatic long-term calculation of time spent in each of two interconnected cages for every individual mouse in a group. In addition, it can be measured how much time individual mice spend together in the same compartment. The determined dwelling time is used to conclude the choice between different enrichment items from the point of view of a mouse. The offered items were categorized and tested by their intended purpose of structuring the cage

(structural enrichment), stimulating foraging engagement of the mice (foraging enrichment), and providing an alternative resting place (housing enrichment). To rank multiple items within those three categories with regard to the preference by the mice, we combined multiple binary choice tests and calculated so-called 'worth values' [31]. In order to further evaluate the quality with regard to consistency of choice among individual mice and within groups of mice living in the same cage we used a recently developed method for analyzing such worth value ratings [32]. The overall aim of assigning worth values to specific enrichment items by multiple comparisons, is to provide scientifically based assistance for improving housing conditions of laboratory mice. A better knowledge of the preferences and the importance of certain elements from the animals' point of view will certainly help to adjust the housing conditions in the laboratories to the animals' needs and thus increase animal welfare.

## Results

### Preference testing

By combining multiple binary choice tests, a worth value is calculated for each item indicating the relative probability of preferences (worth values, ranging from 0 to 1). In order to indicate the amount of disagreement of the tested animals on the calculated worth position, the consensus error (CE) is calculated. CEs of all 12 mice for the enrichment items of the categories foraging, structural, and housing during the entire 46-hours testing cycle and during active and inactive time are given in Fig 1. While the CE indicates the in-between agreement on the ranking positions of the items, the ratio of intransitivity informs about the amount of inconclusive rankings on an individual basis.

Of all foraging enrichment items, the lattice ball received the highest worth value (WV) during the 46-hour testing interval (mean WV: 0.51; CE: 29.17%), both during active (WV: 0.48; CE: 33.3%) and inactive time (WV: 0.42; CE: 45.83%). The overall ratio of intransitive choices was at 13.3%.

Over the total time of 46 hours, the highest worth values regarding the structural enrichments were attributed to the rope (WV: 0.24; CE: 45.83%). However, during the active time the second plane (WV: 0.27; CE: 70.83%) was at the highest position on the scale while during the inactive time both, the second plane (WV: 0.25; CE: 75%) and the rope (WV: 0.25; CE: 50%) reached the highest worth values. The overall ratio of intransitive choices was at 28.3%.

Out of the housing enrichments the wooden angle (WV: 0.25; CE: 61.9%) and the floorhouse (WV: 0.25; CE: 58.33%) were attributed with the highest WV over 46 hours. Within the active time the paper house (WV: 0.30; CE: 54.17%) had the highest WV. During the inactive time all items achieved very similar WV with the floorhouse (WV: 0.21; CE: 79.17%) and the houseball (WV: 0.21; CE: 79.17%) equally ranked in first position. The overall ratio of intransitive choices was at 25%.

Fig 2 illustrates the relative preferences (worth values) of the mice of Group 1 (n = 4), Group 2 (n = 4) and Group 3 (n = 4) for the enrichment items of the categories foraging, structural and housing during the entire 46-hours testing cycle.

Within the foraging enrichments group 1 ranked the lattice ball (WV: 0.26) and the tube with stones (WV: 0.26) on the first position, whereas group 2 and 3 ranked solely the latticeball (group 2 WV: 0.26; group 3 WV: 0.33) on the first position.

Among the structural enrichments group 1 ranked the rope (WV: 0.26) and the second plane (WV: 0.26) on the first position, group 2 ranked the rope (WV: 0.27) first and group 3 ranked the clip with the plastic tube (WV:0.34) first.

Analyzing the ranking positions of the housing enrichments on group level, group 1 ranked the floorhouse and the wooden angle (both WV: 0.26), group 2 the floorhouse (WV:0.33) and

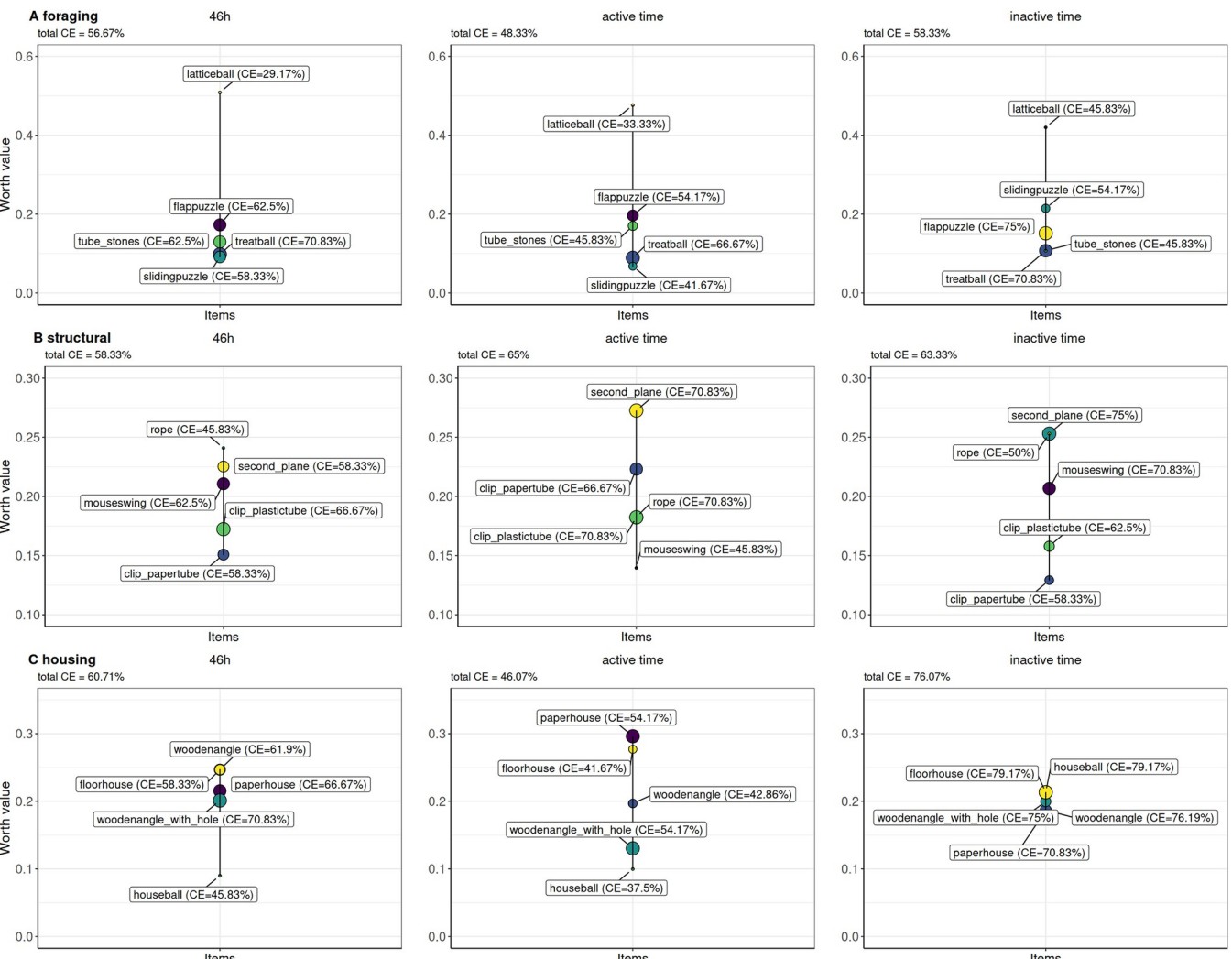

**Fig 1. The relative probability of preferences (worth values) and consensus errors (in percent) of all mice (n = 12) for the tested enrichment items from the categories foraging, structural and housing in the single paired comparisons.** The 46-hour period depicts the complete testing cycle whereas the active time depicts the dark phase of the testing cycle and the inactive time depicts the light phase of the testing cycle.

group 3 the wooden angle, the paper house, and the wooden angle with hole (all three WV:0.25) on the first position.

## Mutual influence within the groups

To test whether the mice in each cage influenced each other's preference, we analyzed the transitions between cages that occurred in temporal proximity within 1 second. Table 1 presents the results of the follow events, the influence events and the proportion of follow events and influence events of the transitions per mouse.

The mean proportion of follow events in the transitions was 1.39% and the proportion of influence events in the transitions was 1.31%. If the interval of temporal proximity of events was increased to 3 s, the proportion of follow events increased to 4.73%.

We also analyzed the time course of the co-location of each possible pair of mice in the same compartment (see S1 Fig). Overall, all mice spent a mean of 70.32% (group 1: 69.63%,

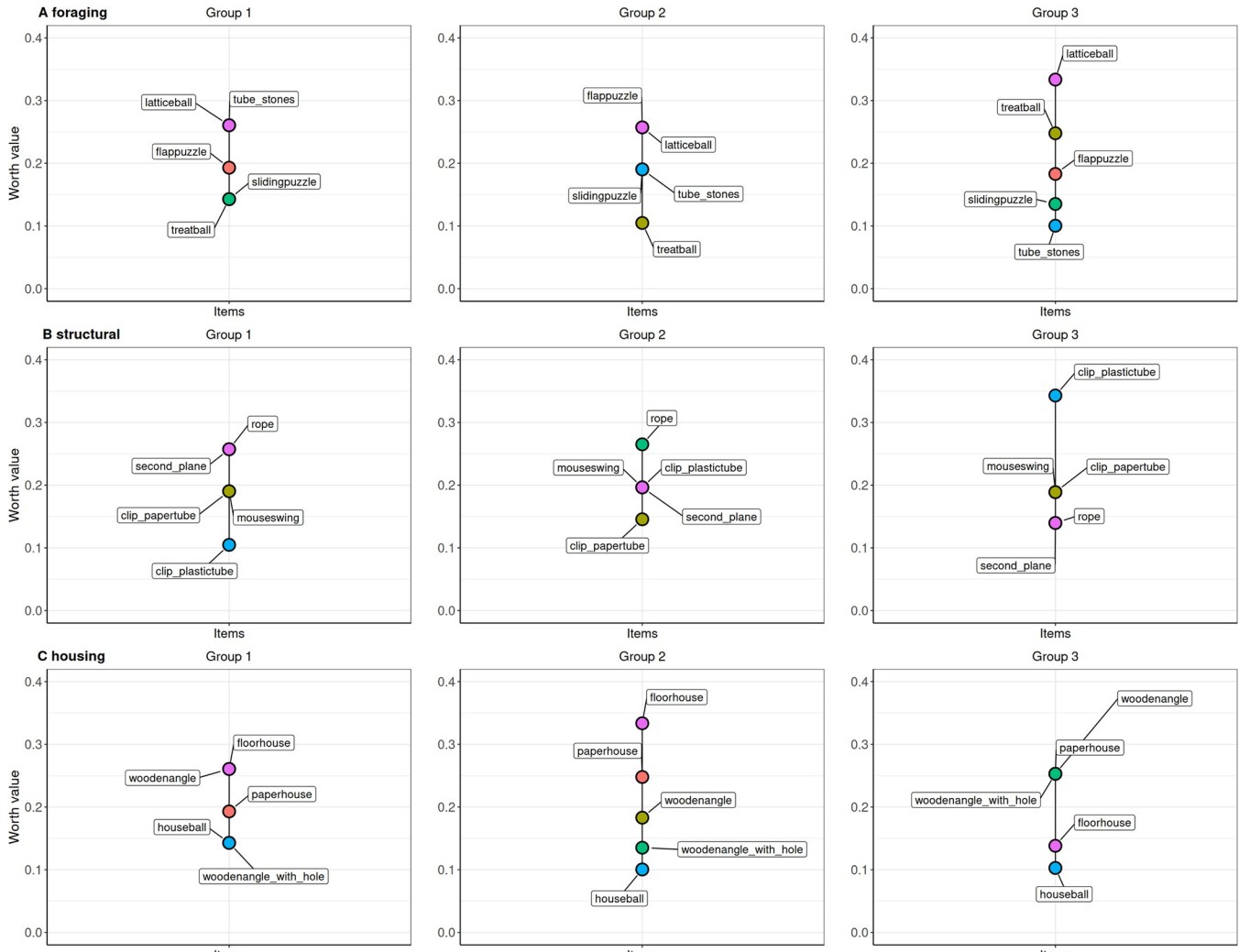

**Fig 2. The relative probability preferences (worth values) of the mice from Group 1 (n = 4), Group 2 (n = 4) and Group 3 (n = 4) for the tested enrichment items from the categories foraging, structural and housing in the single paired comparisons over the entire 46-hour testing cycle.**

**Table 1. The results from the follow and influence behavior analysis of 12 mice from the three experimental groups (1,2,3) of the complete data set.**

| Group | Mouse | Transitions | Follow Events | Follow % | Influence Events | Influence % |
|-------|-------|-------------|---------------|----------|------------------|-------------|
| 1 | 1 | 11608 | 69 | 0.594 | 68 | 0.568 |
| 1 | 2 | 11132 | 54 | 0.485 | 73 | 0.656 |
| 1 | 3 | 10919 | 77 | 0.705 | 58 | 0.531 |
| 1 | 4 | 9955 | 58 | 0.583 | 59 | 0.593 |
| 2 | 1 | 10387 | 229 | 2.205 | 183 | 1.762 |
| 2 | 2 | 10224 | 199 | 1.946 | 193 | 1.888 |
| 2 | 3 | 8442 | 207 | 2.452 | 213 | 2.523 |
| 2 | 4 | 7557 | 143 | 1.892 | 189 | 2.501 |
| 3 | 1 | 7480 | 93 | 1.243 | 81 | 1.083 |
| 3 | 2 | 8440 | 97 | 1.149 | 96 | 1.137 |
| 3 | 3 | 7428 | 88 | 1.185 | 89 | 1.198 |
| 3 | 4 | 6013 | 64 | 1.064 | 76 | 1.264 |

group 2: 70.73%, group 3: 70.59%) of the time together with each conspecific. There is no indication that any of the possible pairs avoid each other.

## Discussion

The aim of this study was to evaluate enrichment elements from the perspective of group housed female C57BL/6J mice. In a series of home cage based binary preference tests, mice could choose between different enrichment elements and their choice behavior was evaluated descriptively. The items were ranked according to a worth value that was calculated from combining the binary choices. In addition, the degree of disagreement in item selection between mice was measured as consensus error (CE) and the percentage of intransitive rankings (i.e., A>B>C>A).

All choice tests were performed while the mice were in their respective social group in one out of three test set-ups with four mice each. It is reasonable to assume that individuals within a social group can influence each other. Overall, all mice switched back and forth between the two cages very frequently. This shows that the items tested were sufficiently familiar and that switching between the cages can be considered an active choice. A closer look at the switches in close temporal intervals allows us to assess how strong the mutual influence on the choice decision was. Therefore, we conducted an analysis of follow and influence behavior, which shows how attached individual choice is to decisions of conspecifics. Data revealed that the three groups indeed did not come to the same conclusion with regard to choosing preferred items. However, there was no considerable attraction to individual mice that could explain the respective preference as a trend triggered by individual influencer mice. Overall, a mean follow rate of 1.39% is reflecting a negligible direct immediate impact on individual choices. Even if a longer, more conservative follow interval of 3 seconds was applied, more than 95% of all cage changes were not directly related to an influencer. Thereby we could demonstrate that group housed mice can explore a choice test apparatus without being directly led by others and thus an independent choice of location is likely. Even though the mice had the opportunity to freely decide for or against the cage with the tested items, other factors may have been more important for the decision and must be considered. For example, the preference for a certain cage may also depend on scent marks, or the spatial distribution of the group. It can be assumed that the shared nest is usually also located in the preferred cage. Once the nest is built, it could act as an amplifier and thus be a strong predictor of group choice, especially during the inactive phase. Overall, all groups showed a strong tendency to spend most of their time together, which underlines the strong social cohesion that certainly plays a role in the choice of where to stay, even if no direct influence is obvious. Nevertheless, our data reveals that those elements that were preferred more unequivocally (e.g., the lattice ball) led to an overall peak in time spent together while on days where more ambiguous elements were tested the time spent together decreased (see S1 Fig). Male and female mice behave differently and therefore sex may also have an influence on choice behavior in mice. However, with regard to housing conditions previous research so far has shown that sex differences are negligible in preference studies with mice [4,26,28,33]. However, it is worth noting that male mice are often housed singly in order to avoid aggressive behavior. Here we solely investigated preference in female mice, so no conclusion can be drawn on the influence of sex and group housing was chosen as the most representative social housing condition for female mice. Nevertheless, testing groups of mice will remain a challenging issue especially with regard to choosing the correct statistical unit [29,34,35]. On the other hand, mice are social animals in nature, and in accordance to underlying legislation [1], single housing should be avoided under experimental conditions if possible. Furthermore, it is arguable whether choice decisions of individually kept animals can

be transferred one-to-one to animals in a social group resembling realistic laboratory conditions [29]. Future studies with individually housed mice could add important data to improve housing conditions for single housed animals, which may be unavoidable under some experimental constraints. With regard to ranking the items, we argue, if the strength of individual preference is so small that it is overridden by the peer pressure of the social group that this is probably an indication of only a small difference between the valence of the two items. Thus, we decided against testing individual animals and used the option of the home cage based choice experiment to study the mice as socially living animals within the group [36]. Furthermore, in addition to analyzing the results of all mice over the total test duration of 46 h, we subdivided the results into an active phase (dark) and inactive phase (light) of the mice [4,37]. This served to evaluate possible preferences associated with active (e.g., climbing, gnawing) or inactive (e.g., sleeping, resting) behaviors of the enrichment items by the mice.

To investigate whether the mice agreed in their choice of preference, we calculated a consensus error (CE) to display the amount of disagreement. Since this measure has only recently been introduced for scaling preferences [32] it is probably not easily interpreted intuitively. Generally speaking, low CE scores indicated a high agreement, whereas high scores reflect a low agreement. The CE is calculated for each item, based on the comparison with all other items and gives the mean percentage of disagreement regarding the ranked position of the respective item. For example, a CE of 50% means that on average 9 out of 12 animals agreed in their preference regarding that specific item.

Overall, the evaluation at the level of all mice of the three groups revealed a high average CE in all analyses and thus a lower agreement in choice, indicating different perceptions of enrichment within a group of mice. The individual group analysis showed that the rank positions of the tested enrichment elements sometimes varied greatly within their categories, resulting in a high overall CE. However, our assessment of follow and influence rates showed that this cannot be explained easily by dominance and following behavior. Therefore, the social dynamics underlying choice within a group are deemed to be more complex. In addition, the test items were freely available through the preference test, so the mice may not have perceived the test as forcing them to choose one or the other. We suggest that the difference in valence between the presented enrichment items may not have been large enough to provoke an unequivocal choice. This is also indicated by the percentage of intransitive rankings which show that individual mice did not always have a clear linear ranking when combining all data. Indeed, it has been shown that the CE and ratio of intransitivity is larger in rankings with low valence ranges compared to large valence ranges in the data provided with the R-package SimsalRbim [32]. Therefore we assume that the differences of valences within the categories were not sufficient to come to unequivocal rankings.

Foraging enrichments were ranked with closely spaced worth values in all assessments. Only the lattice ball stands out with a high worth value, both at the group level and at the overall level. This is also reflected in the CE, which was the lowest in all calculations for the entire period at 29.17% (CE in 46 h of all mice; roughly speaking the CE indicates that on average 10 out of 12 mice preferred the lattice ball). Unlike the other enrichment items in the same category, the lattice ball was attached to the cage top using a metal ball chain, while the tube with stones, the flappuzzle, the slidingpuzzle, and the treatball were placed on the floor, resulting in high visual and functional differences. Due to the fact that after pulling paper out of the ball and eating the millet, the mice were still able to interact with the ball as a moving object to gnaw at or to climb on, it might have been more interesting and hence preferred to interact with. In addition, during the cleaning process, we observed that mice used the extracted paper strips from the lattice ball as additional nesting material. In fact, it has already been established that the combination of different materials for nest building is common in mice [28,38].

Indeed, nesting material is highly valued by mice [33,39] and the motivation to build nests is strong [40,41]. The direct association of the lattice ball with supplementary nesting material may explain the preference for this side of the testing system also during the inactive phase of the mice. Our parallel study investigating the use of the lattice ball in the home cage showed that the active interaction with this design element was less frequent than with other foraging elements of the same category during the active phase [42]. However, in that study the use was evaluated by direct observation during a 30-minute period in the presence of other enrichment elements from other categories. In the present study the data was obtained over two circadian cycles in a binary choice test which might be more conclusive with regard to the overall attractiveness. The elements 'treatball', 'slidingpuzzle', 'flappuzzle', and the 'tube filled with stones' led to inconsistent scores in the worth values and thus low ranking positions in the evaluation at individual group and overall level. Nevertheless, they might serve as cognitive stimulation for mice and enable natural behaviors like burrowing and foraging. This is especially true when considering the high active usage while the elements were filled with millet seeds as additional treats.

Structural elements did also not reveal a complete unequivocally ranking which is again indicated by high values for the CE. However, within this category the second plane and the rope were highly ranked during both the active and inactive time of the mice. The second plane serves as a climbing element as well as for gnawing and as a refuge and sleeping place. The multifunctionality offers a wider range of possibilities for interaction compared to simpler climbing enrichments (i.e., mouseswing, clip with paper or plastic tube, rope). Our previous study found a high rate of second plane-use by mice in video analyses, supporting this hypothesis [42]. Leach et al. [43] also acknowledged a platform-like insert for mouse cages as an appealing enrichment element for mice with its dual function as a resting place and as an object that encourages exploration, jumping, and hiding. In addition, we observed that mice frequently built their nests under the second plane, both during the previous housing period as well as under the test conditions. The other structural enrichment, which was also ranked in high position, was the rope. In contrast, the evaluation of short-term usage in a previous study revealed this item, along with other climbing enrichments that were fixed at the cage top, to be less used when it was presented in a combination of enrichments [42]. The rope was made of hemp and similar to the paper strips derived from the lattice ball, fragments of gnawed hemp ropes were used as additional nesting material. Therefore, the known attractiveness of nesting material [33,39] and the strong motivation to build nests [40,41] might explain the high rank of the cage containing the rope. This again shows that long-term observations are helpful to obtain more conclusive information about the overall attractiveness of the respective enrichment elements. Gjendal et al. [44] found hemp ropes to strengthen the participation in social behavior and encouraging climbing and gnawing behavior in male mice without adverse effects on anxiety levels, stress and aggressive behavior. Hemp ropes can therefore be applied as a simple and inexpensive enrichment for mice and serve as climbing, gnawing, and supplemental nesting material.

Housing enrichment worth values were closely spaced, with apparent differences between groups, and elements partially achieving a reversed ranking. Accordingly, the CEs were considerably high for the overall rating of housing enrichment. Especially in the inactive time, CE ranging from 71% to 79% indicate that only 7 to 8 out of 12 mice had a similar preference on average. Interestingly, van Loo et al. [26] found a paperhousing comparable to the one we used to be preferred over a triangular plastic house. Therefore, we expected the paperhouse to be valued highly more consistently, however, this could not be confirmed unequivocally in our preference tests for all individual groups. Nonetheless, the paperhouse achieved the second place rank during the total and first place rank during the active time in the overall ranking.

Indeed, a video observation revealed the frequent use of the paperhouse during the active phase of the mice [42]. Apparently, the lightweight and easily manipulated structure makes the paperhouse attractive as a movable and changeable object with which the home cage can be actively configured. The floorhouse was also rated highly and seems to promote behaviors such as climbing, hiding and exploring more strongly. Due to its platform-like structure, it also offers a larger additional surface area for these types of behavior. Conversely, the houseball provides the least surface and was ranked to the lowest positions overall and during active time. During the inactive phase, no housing enrichment achieved a clear preference and all houses ranked closely spaced. This suggests that even though nest boxes are perceived as important exploratory objects for mice during the active phase rather than just a mere refuge and sleeping area [33,45], there is little difference in use during the inactive period. It also shows that when mice are asked about their preference for provided items, the answer may be based on a different way of using these items than was expected by the experimenter. It is already established that mice prefer a cage with a nest box to a cage without a nest box [45]. Provision of nest boxes and nesting material increases animal welfare without negatively impacting data variability [14] and therefore, should be provided as an essential enrichment for mice [3]. Since the nest box serves more than just shelter, the choice of design should also take into account the activity-promoting effect of the housing enrichment. Therefore, factors such as additional space or the changeable structure make the floorhouse and paperhouse recommendable.

To determine the effectiveness of enrichment items, it is essential not only to conduct preference tests, but also to examine the ways in which enrichment items are used [20]. Evaluation of the type and amount of interaction via behavioral analysis is therefore deemed an important component to create more species-appropriate housing conditions for mice [42]. Although we cannot provide a statement about the motivational strength [4,22,46], the experimental design used here allows ranking of the different design elements. Determination of motivational strength to obtain resources can be achieved through consumer demand tests and represents the price, for example in form of lever presses or nosepokes [4,47] that an individual is willing to pay for access to certain enrichment elements [4,19,23]. Nevertheless, our study shows that when mice have a say, judgments about a reasonable type of enrichment can be made in a somewhat more evidence-based manner.

The age of the mice used in our study was approximately one year, which is generally an unusually old age for mice used in biomedical research. Data on the influence of age on preferences for enrichment is sparse, for example, van Loo et al, 2004 [48] found that preference in male mice for nest material was high at all ages, but preference for social contact increased with age. Although we would not expect dramatic changes in preference for the presented enrichment items with age, our results may be especially relevant for older mice, such as those used in Alzheimer's research or research on aging processes.

Overall, the high CEs in our study, especially for housing and structural enrichment, reflects individual differences in the assessment of the different enrichment elements from the perspective of each mouse. It should also be borne in mind that objects that are very similar cannot always be clearly distinguished from each other in terms of their valence [32]. However, the fact that not all animals have always made a clear choice does not in any way indicate in principle that enrichment is superfluous. On the contrary, a comprehensive body of literature [2–4,6–8,10–15,20,49] shows positive effects of enrichment. From our study, in addition to practical recommendations, we can also derive the possibility of using different enrichment elements as a means of variation without depriving the mice of any one enrichment item that they would desperately want.

Indeed, to create an interesting and stimulating living environment for mice, it is important to provide variety through regular exchanges. Varied housing can help prevent behavioral

deprivation [50,51] and behavioral disorders [52] in laboratory animals by enabling species-specific behaviors. Furthermore boredom in laboratory animals [53,54] in its severe chronic forms shares symptoms with learned helplessness and depression and should therefore be treated as an important animal welfare concern [55]. There might be concerns that improving housing conditions lead to changes of brain and behavior and thus data derived from these models being less comparable to literature data based upon impoverished housing. We argue that data that could only be reproduced under the same impoverished conditions would severely lack external validity and thus be scientifically doubtful [56]. In translational research it is aimed to infer conclusions from animal models to human conditions; any lack of normal behavioral development of the model species therefore must be seen as a factor which worsens the translational value. Therefore it should be the ambition of every experimenter to improve the well-being of laboratory animals and thus enhance the quality of animal experiments [57]. Legal husbandry regulations [1] should indeed be considered as a minimum requirement that does not place an upper limit on the genuine improvement of living conditions of laboratory animals [21].

## Conclusions

In our study, preferences for different enrichment items were evaluated in female C57BL/6J mice using a home cage based preference test system. This easy-to-use method for translating binary preferences into scaling a number of enrichment items according to calculated worth values facilitates decisions on the use of enrichment in laboratory husbandry. As foraging enrichment, the lattice ball with its multifunctional character of activity stimulation and its content of paper strips as additional nesting material achieved high worth values. A rope made of hemp also achieved high position in the worth scale and serves as a structural element for climbing, gnawing, and providing additional nesting material. As a structural enhancement, a second level of wood, used both as a resting place and for active engagement, was at the top of the ranking. No clear preferences were found for the type of housing during the inactive period of the mice. This indicates that all housings presented for selection were basically suitable as sleeping places.

High CEs within the studied rankings suggest a strong individuality in the perception of the enrichment elements. Therefore, a multifaceted enrichment approach should be considered to meet the needs of individual mice. Increasing the complexity of housing for laboratory mice toward a more stimulating environment allows them to exhibit a more species-specific behavioral repertoire, potentially leading to more reliable animal models in biomedical research.

## Materials and methods

### Ethical approval

All experiments were approved by the Berlin state authority, Landesamt für Gesundheit und Soziales, under license No. G 0069/18 and were in accordance with the German Animal Protection Law (TierSchG, TierSchVersV). The study was pre-registered in the Animal Study Registry (ASR, DOI 10.17590/asr.0000162) and is reported in accordance with ARRIVE guidelines (https://arriveguidelines.org).

### Animals and housing condition

Twelve female C57BL/6J mice were purchased from Charles River Laboratories, Research Models and Services, Germany GmbH (Sulzfeld). The sample size was chosen to ensure a statistical power of 80% and an alpha value of 0.05. Due to the exploratory experimental

approach, the effect size is unknown and had to be estimated on the basis of published studies with comparable experimental designs as well as own experiments from our laboratory. The mice were 7–8 weeks of age upon arrival in the animal facilities. Mice were randomly allocated to groups of four animals in Makrolon type III cages by a researcher not involved in the experiment; animals were alternately assigned to the groups (1,2,3) to avoid bias. During the first three weeks the animals were housed in groups of four animals in type III Makrolon cages (L x W x H: 425 x 265 x 150 mm, Tecniplast, Italy) with aspen bedding material (Polar Granulate 2–3 MM, Altromin), paper (cellulose paper unbleached 20x20 cm, Lohmann & Rauscher International GmbH & CO KG) and cotton roll nesting material (dental cotton roll size 3, MED-COMFORT), a 15 cm transparent plexiglas tube (Ø 4cm PMMA xt®, Gehr®) and a red triangle plastic house (mouse house, TECNIPLAST®). They were provided with regular rodent food (autoclaved pellet diet, LAS QCDiet, Rod 16, Lasvendi, Germany) and tap water *ad libitum*. Room temperature was maintained at 22°C (+/- 2), room humidity at 55% (+/- 15) and a 12/12 light/dark cycle regimen (lights off 20:00) with simulated sunrise between 7:30 and 8:00 using a Wake-up light (HF3510, Philips, Germany). To further implement refinement procedures according to the 3Rs [58] all mice were trained to tunnel handling [59] daily during the habituation phase and tunnel handling was used throughout the whole experiment.

At the age of eleven weeks mice were provided with cage enrichment. Cages were cleaned weekly and each mouse was subjected to a visual health check. The enrichment scheme consisted of permanently provided items (running disc with mouse igloo, paper nesting, cotton rolls, Table 2) and five weekly rotating items from structural, housing, nesting and foraging categories (See Tables 2 and 3). These enrichment items were randomly exchanged during the weekly cage cleaning. Randomization of the enrichment combination was done with the use of the function randomize() in the software R (version 4.0.4). To motivate the mice in solving the riddles of the foraging enrichment category, a small amount of millet seeds was provided in the morning inside the riddle during the daily animal inspection. Prior to the preference experiments, the mice were used in another behavioral experiment [42] to observe the usage of the enrichment items but they stayed in the above-mentioned housing conditions.

### Animal identification

For individual animal identification, all animals were provided with a RFID transponder (ISO 11784/85, FDX-B transponders, Planet ID®) under the skin of the dorsal neck region in rostro caudal implantation direction. This procedure took place at the age of 9–10 weeks under general isoflurane anesthesia and pain reliever (Metacam®).

Additionally, all mice were color-coded weekly on the tail with a permanent marker (Edding® 750) to easily distinguish them in daily visual and weekly health checks.

### Preference testing

After 43 weeks in the enriched housing condition, preference tests were conducted using the Mouse Positioning and Surveillance System (MoPSS) [29]. The system consisted of two macrolon type III cages, connected with a 30 cm plexiglas tube. Two RFID antennas were attached outside the tube. Inside the tube, plastic barriers were installed in order to slow down mouse movement (see Fig 3). The RFID antennas were connected to a reader, which recorded the mouse movements between the left and right cage through detection of the implanted RFID transponder.

The mice remained in their group of four animals, and three preference systems were used in parallel. The systems were positioned in a row on a steel table, in an experimental room with the same environmental conditions as during the housing period.

**Table 2. Used enrichment items.** Permanently available enrichment items used during housing period prior to or during MoPSS preference testing.

| deployment | enrichment item | |
|---|---|---|
| **standard house in MoPSS experiment** | **triangular house**<br>(mouse house, TECNIPLAST®) |  |
| **housing used in housing period** | **running wheel**<br>(fast-trac + mouse igloo, Bio-Serv®) |  |
| **permanently available (housing and MoPSS experiment)** | **paper nesting**<br>(cellulose paper unbleached 20x20 cm, Lohmann & Rauscher International GmbH & CO KG) |  |
| | **cotton roll**<br>(dental cotton roll size 3, MED-COMFORT) |  |
| **nesting used in housing period in home cage** | **fine wood wool**<br>(H0234-NBF, ABEDD®) |  |
| | **coarse wood wool**<br>(H0234-NBU, ABEDD®) |  |
| | **square hemp pads**<br>(H3279-10 eco- hemp, ssniff Spezialdiäten GmbH) |  |
| | **folded paper strips**<br>(sizzlenest®, datesand Ltd) |  |
| | **mid coarse wood wool**<br>(NBGE012, ABEDD®) |  |

**Table 3. Tested enrichment items.**

| category | enrichment item | |
|---|---|---|
| housing | **houseball**<br>(crawlball, Bio-Serv®) | |
| | **floorhouse**<br>(safe harbor, Bio-Serv®) | |
| | **paperhouse**<br>(LBS Serving Biotechnology) | |
| | **wooden angle**<br>(climbing roof, ABEDD®) | |
| | **holed wooden angle**<br>(holed climbing roof, ABEDD®) | |
| structural | **second plane, 1 hole**<br>(1 hole lying boards for cage type III, ABEDD®) | |
| | **second plane, 2 holes**<br>(2 hole lying boards for cage type III, ABEDD®) | |
| | **clip with paper tube**<br>(38 x 1.25 x 75 mm play tunnel and tunnel clip, Datesand Ltd) | |
| | **clip with plastic tube**<br>(Plexiglas tube transparent 70mm Ø, KUS and tunnel clip, Datesand Ltd) | |
| | **mouseswing**<br>(single mouse swing, Datesand Ltd) | |
| | **mouseswing double**<br>(double mouse swing, Datesand Ltd) | |
| | **rope**<br>(jute yarn 6-ply, 6mm, Rayher 4200531) | |

*(Continued)*

**Table 3.** (Continued)

| category | enrichment item | |
|---|---|---|
| foraging | **treatball**<br>(self-designed and printed with Filamentworld schneeweiß PLA, https://tinyurl.com/3dtreatball) | |
| | **slidingpuzzle**<br>(Interactive Smart Toy, Living World® green) | |
| | **tube with stones**<br>(mouse tunnel, Bio-Serv® and white marble pebbles 15–25 mm Ø, Min2C Natural Minerals) | |
| | **lattice ball with metal chain**<br>(Hol-ee Roller® size mini, JW®) | |
| | **flappuzzle**<br>(self-designed and printed with Filamentworld PLA, https://tinyurl.com/3dflappuzzle) | |

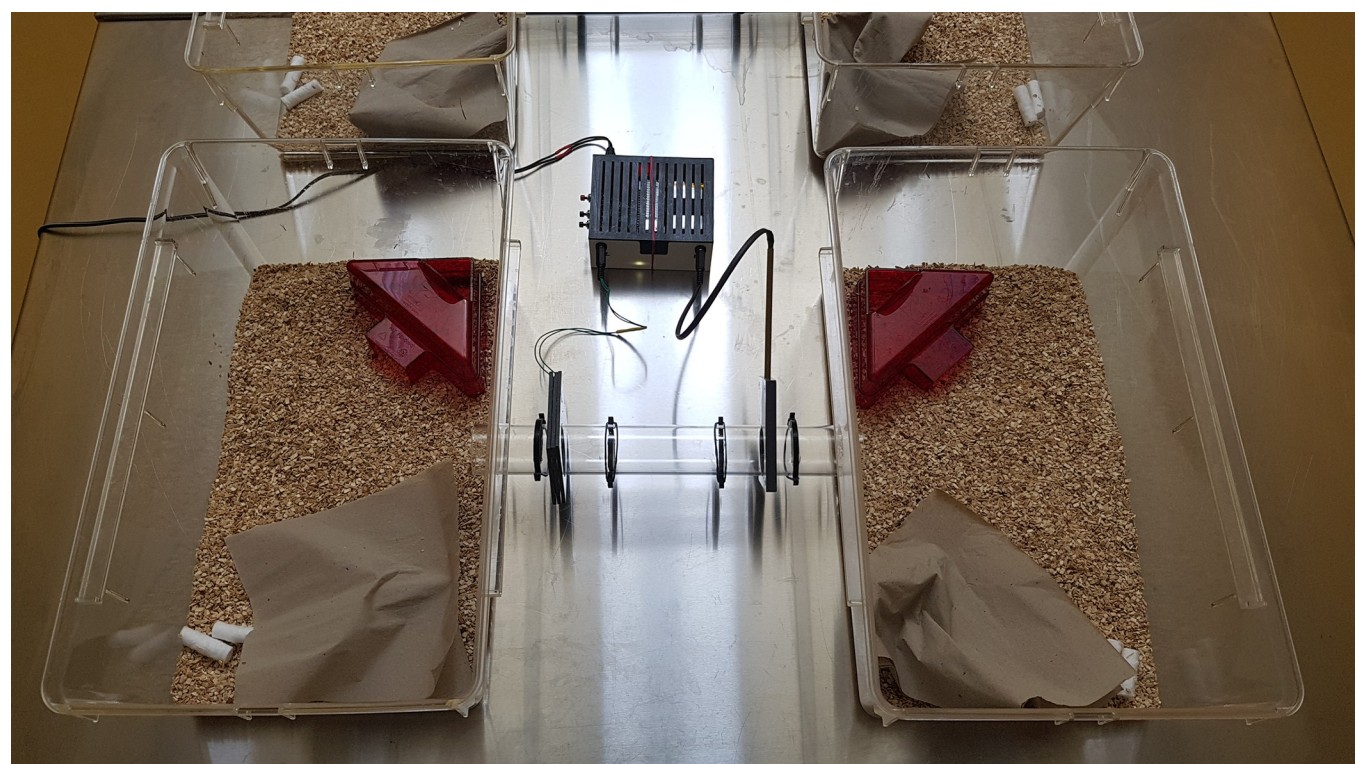

**Fig 3. The Mouse Positioning and Surveillance System (MoPSS).**

To achieve the same lighting conditions for the left and right cage of the preference system, four LED lights (Brennenstuhl® Dinora 5000 Baustrahler 47 W 5000 lm Tageslicht-Weiß 1171580) on tripods were set up pointing towards the ceiling. Light intensity in both test cages was checked with a lux meter (voltcraft® light meter MS-1300).

The testing cages were outfitted with 150 g aspen bedding (Polar Granulate 2–3 MM, Altromin), a red translucent triangular plastic house, three uncolored paper towels, two cotton rolls, and water and rodent food (autoclaved pellet diet, LAS QCDiet, Rod 16, Lasvendi, Germany) *ad libitum* with same amount on each side (see Table 2 for equipment details). Enrichment items placed into the cages were visible so that a full blinded design was not achievable. However, the automated recording of the behavioral data in the absence of the experimenter excluded any possible influence.

Enrichment items of one category each were randomly presented twice for 23 hours starting at 10:00 am until 9:00 am the following day. Comparisons were made of two enrichments from the same category each time. This was done in order to come to results that are interpretable with regard to the prospective utility the items hold within their respective category. For example, during the inactive phase we would expect a clear preference for a cage where a housing enrichment was placed if no housing but an element of the foraging category was placed in the other cage. Therefore comparing only items from the same category helps to avoid misleading conclusions. Between the two sessions using the same items, the enrichment items were switched between the cages to counterbalance possible side preferences. In addition also the nesting material and bedding was mixed between the left and right cage and the mice were supplied with their daily amount of millet seeds. The first category tested was the 'structural enrichments' followed by the 'foraging enrichments' and the 'housing enrichments'. Two days before the first preference test, the mice were introduced to the experimental setup including the MoPSS for habituation. Over a period of 15 weeks, systematic comparisons were thus subsequently made to obtain the data for the preference analysis. After completion of the experiments in this work, the animals remained in their housing conditions and were used for further studies.

## Analysis of preference

The mouse tag detections were automatically saved onto a microSD card during the experiment and each detection was marked by a current timestamp with the antenna number (left/right) and the individual mouse RFID tag number. Data analysis and sanity checks with logical correction of missing detections were done using a data evaluation script in the software R (R version 4.0.4, R Studio version 1.3.959) specially developed for MoPSS data analysis [29]. No missing data were found, all mice were regularly detected and none had to be excluded from analysis of stay times. Stay times for each of the twelve mice in each cage side were calculated as times between cage changes when a mouse tag was detected at a new cage. It has been shown that the time spent in the tube is negligible for preference calculation [29] and therefore we did not subtract the time spent passing the tube. Stay durations over the 46 hours testing period of each single experiment were summed up for each mouse and then calculated as percentage of the total time. Over the entire test period, the mice spent almost 50% (49%-51%) of their stay time on each side of the preference test system, so a dominant side preference could not be found. All data was analyzed both at group/cage level and in relation to the length of stay of all individual mice over the total period of 46 hours and over the light and dark phase representing the activity phases of the mice. The calculated percentages of stay durations were then used for comparison of side preferences (left vs. right cage) for enrichment one and enrichment two including a side switch of the presented items. The raw data with stay durations in percentage during the complete 46 hours testing period and divided into the active and inactive time period can be found in the supplementary material (S1–S3 Tables).

To rank the tested enrichment items regarding the strength of the preference for each item, a method developed by Hatzinger et al. 2012 [31] of combining the multiple single binary choices to a 'worth value' was performed using R and the package simsalRbim [32]. A similar method was used by Hopper et al. 2019 to determine the worth value of different items of food rated by a male gorilla [60]. In short, to estimate the position of an item, the 'worth value' of each enrichment item was calculated based on the prefmod package [31] with its fit to a log-linear Bradley-Terry model (LLBT). The LLBT was specifically made for paired- comparison testing and estimates a subject´s relative 'worth value' for each choice on a probability of preference scale that sums to 1 [31]. Greater probability of preference is represented here by a higher 'worth value'.

To determine the agreement amongst the mice regarding the 'worth value' for each ranked enrichment item and its estimated position on the scale, a consensus error (CE) was also calculated using the simsalRbim package [32]. A detailed example of the calculation of the CE can be found on the simsalRbim homepage [32]. In brief, the CE reflects the extent of agreement that the mice showed regarding the preference for a certain enrichment in binary choices over the other tested enrichment items. A value of 0% points to a perfect agreement of a ranking position and 100% indicates a full disagreement of all individual mice (i.e., 6 mice prefer item A while the other 6 mice prefer item B). It should be noted that CE is biased by the number of individuals, with low numbers resulting in CE being significantly more affected by a single animal. In our presentation of the cage wise preferences we therefore refrained from calculating the CE as the ratings are based on a choice of only 4 animals. As a second measure of quality of the determined rankings the percentage of intransitivity was calculated. In brief, an intransitive triplets is an indecisive ranking of three elements with A>B>C>A. In a ranking with 5 elements, these elements can be divided into 10 possible triplets. Intransitive choices are measured per individual and therefore with 12 mice there are overall 120 triplets in the rankings for each category that can be either transitive (consistent in their order) or intransitive. The ratio of intransitivity is calculated as the percentage of intransitive triplets from all 120 possible triplets using the simsalRbim package [32]. A high percentage of such intransitive triplets indicates that there is no clear individual consistency regarding the ranking of the items.

All analyses were run in R version 4.0.4 using RStudio (Version 1.3.959).

## Sample size

It is debated whether or not group housed animals can be unequivocally considered to act independently in their choice and therefore each cage would have to be considered as one independent sample [29,34,35,61]. This presents a dilemma because the mice would either have to be housed individually or the total number of experimental animals would have to be increased by the use of additional cages. As we are explicitly interested in the preference for enrichment items under common social conditions, housing mice singly was not an option. With regard to keeping the overall number of experimental animals as low as possible in the light of the 3Rs, we calculated that 12 mice would be a reasonable sample size if they indeed act independently. In order to demonstrate that individual preference was an independent choice, we conducted a follow and influence behavior analysis using R (Version 4.0.4) with our obtained experimental data from the MoPSS. A follow event was defined as a transition of one mouse directly detected within one second after another mouse. The leading mouse detected in this constellation received an influencer event. We further calculated a follow rate and influence rate as follows:

$$follow\ rate = \frac{follow\ events}{total\ transitions}$$

$$influence\ rate = \frac{influence\ events}{total\ transitions}$$

In addition, we analyzed the time each possible pair of two mice spent in the same cage over the course of the study (S1 Fig). While a strong preference for a certain cage (or any given resource therein) will lead to an overall increase of the time all pairs spend together, this analysis especially allows to detect if certain individuals avoid each other. Such avoidance might indicate, for example, a dominance hierarchy in which one individual monopolizes a resource.

## Supporting information

**S1 Fig. The experimental time (days 1–60) in percent spent by the mice within the same compartment of the MoPSS test system (left and right cage) divided by Group 1 (n = 4), 2 (n = 4) and 3 (n = 4) and observational period (active time, inactive time, total time).** Mice were displayed as couples (m1-m2: mouse 1 and mouse 2, m1-m3: mouse 1 and mouse 3, m1-m4: mouse 1 and mouse 4, m2-m3: mouse 2 and mouse 3, m2-m4: mouse 2 and mouse 4, m3-m4: mouse 3 and mouse 4).
(TIF)

**S1 Table. Raw data with stay durations in percentage of the 12 mice during the whole 46 hours.**
(PDF)

**S2 Table. Raw data with stay durations in percentage of the 12 mice during the active time period.**
(PDF)

**S3 Table. Raw data with stay durations in percentage of the 12 mice during the inactive time period.**
(PDF)

## Acknowledgments

The authors thank the animal caretakers of the Federal Institute of risk assessment, especially Carola Schwarck and Lisa Gordijenko, for their support in the animal husbandry.

## Author Contributions

**Conceptualization:** Ute Hobbiesiefken, Lars Lewejohann.

**Data curation:** Ute Hobbiesiefken.

**Formal analysis:** Ute Hobbiesiefken.

**Funding acquisition:** Lars Lewejohann.

**Investigation:** Ute Hobbiesiefken.

**Methodology:** Ute Hobbiesiefken.

**Project administration:** Kai Diederich, Lars Lewejohann.

**Software:** Birk Urmersbach, Anne Jaap.

**Supervision:** Kai Diederich, Lars Lewejohann.

**Visualization:** Ute Hobbiesiefken.

**Writing – original draft:** Ute Hobbiesiefken.

**Writing – review & editing:** Birk Urmersbach, Anne Jaap, Kai Diederich, Lars Lewejohann.

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
