## [Decision Letter · Decision Letter 0]

13 Sep 2022

PONE-D-22-23528Rating enrichment items by female group-housed laboratory mice in multiple binary choice tests using an RFID-based tracking systemPLOS ONE

Dear Dr. Lewejohann,

Thank you for submitting your manuscript to PLOS ONE. After careful consideration, we feel that it has merit but does not fully meet PLOS ONE’s publication criteria as it currently stands. Therefore, we invite you to submit a revised version of the manuscript that addresses the points raised during the review process.

The three reviewers clearly identify a number substantial points that need to be addressed. They include not only rewriting parts of the manuscript, but also reanalysis. Please carefully respond to each critique.

If you feel that you address all concerns, please submit your revised manuscript by Oct 28 2022 11:59PM. If you will need more time than this to complete your revisions, please reply to this message or contact the journal office at plosone@plos.org. Please include the following items when submitting your revised manuscript:A rebuttal letter that responds to each point raised by the academic editor and reviewer(s). You should upload this letter as a separate file labeled 'Response to Reviewers'.A marked-up copy of your manuscript that highlights changes made to the original version. You should upload this as a separate file labeled 'Revised Manuscript with Track Changes'.An unmarked version of your revised paper without tracked changes. You should upload this as a separate file labeled 'Manuscript'.If applicable, we recommend that you deposit your laboratory protocols in protocols.io to enhance the reproducibility of your results. Protocols.io assigns your protocol its own identifier (DOI) so that it can be cited independently in the future. For instructions see: https://journals.plos.org/plosone/s/submission-guidelines#loc-laboratory-protocols. Additionally, PLOS ONE offers an option for publishing peer-reviewed Lab Protocol articles, which describe protocols hosted on protocols.io. Read more information on sharing protocols at https://plos.org/protocols?utm_medium=editorial-email&utm_source=authorletters&utm_campaign=protocols.

We look forward to receiving your revised manuscript.

Kind regards,

Andrey E Ryabinin, Ph.D.

Academic Editor

PLOS ONE

Journal Requirements:

"The authors received no specific funding for this work. The work is financed by the annual budget of The German Federal Institute for Risk Assessment (BfR). BfR reports to the Federal Ministry of Food and Agriculture (BMEL). The BMEL had no role in study design, data collection and analysis, decision to publish, or preparation of the manuscript."

"This study was funded by the Federal Institute for Risk Assessment, Max-Dohrn-Straße 8-10, 10589 Berlin, Germany. The funders had no role in study design, data collection and analysis, decision to publish, or preparation of the manuscript."

Reviewers' comments:

Reviewer's Responses to Questions

**Comments to the Author**

1. Is the manuscript technically sound, and do the data support the conclusions?

Reviewer #1: Yes

Reviewer #2: Yes

Reviewer #3: Yes

2. Has the statistical analysis been performed appropriately and rigorously? 

Reviewer #1: Yes

Reviewer #2: Yes

Reviewer #3: No

3. Have the authors made all data underlying the findings in their manuscript fully available?

Reviewer #1: Yes

Reviewer #2: Yes

Reviewer #3: Yes

4. Is the manuscript presented in an intelligible fashion and written in standard English?

Reviewer #1: Yes

Reviewer #2: Yes

Reviewer #3: Yes

5. Review Comments to the Author

Reviewer #1: This article is simple in concept and design but the authors thoroughly and elegantly describe the justification of their work and the results. I have minor comments that I think should be addressed in the discussion section.

1) The authors state that preference tests were performed comparing items from different "categories" of enrichment, but all the comparisons presented show preferences within categories. Did the mice have consistently any preference for one category over another? This could actually be worth making a figure ranking the categories in each group to visualize any effects (or lack thereof).

2) This study was performed exclusively in female mice. The authors cite an article investigating enrichment in male mice, but do not compare their results in the discussion section. With the importance of sex as a biological variable I think this is worth discussing, especially since I cannot seem to find access to the cited article studying male mice.

3) It may be worth pointing out the age of the mice in the discussion section. These mice are about a year old, which I would consider to be significantly older than the age of subjects used in most studies. How might these results compare to mice closer to a more common experimental age (e.g. 2-4 months old)? Could this question of enrichment quality, selection, and variation be of particular relevance to studies making use of older animals (e.g. aging studies, Alzheimer's research)?

Reviewer #2: The manuscript of Hobbiesiefken et al. examines how mice housed a group setting show individual preferences for particular structural, housing, and foraging enrichment objects. Using ID tagging of each mouse in combination with a two connected home-cage based assay, they track the preferences of individual mice over a 46 hour period and show that individual mice as well as groups show varying preferences for objects with no clear preference across the whole cohort. On a whole, I thought the study addresses a useful consideration in behavioral neuroscience, was interesting to read, and does a good job making use of ID tagging technologies to examine how mice influence each other's behavior in the homecage.

I have a few questions and suggestions which I feel would strengthen the conclusions of the manuscript:

1. My main concern is that the data about individual animal preferences may be heavily confounded by time spent in one object's chamber but not directly interacting with the object. As the authors noted, time spent on one side or the other can be biased by the presence of a nest, for example, where the animals may choose to sleep or rest for long durations of time. Morever, some animals may innately be less exploratory than others, and may spend more time in the periphery of the cage rearing or resting even when awake. As such, I think the authors ought to analyze the data in a more precise manner by using the tracking data to identify bouts when the animals are directly interacting with the object, for example by thresholding for frame in which the animal is within a certain distance of the object. The authors could also complement their RFID tracking with multi-animal pose estimation to do this if necessary, as I'm not totally sure if RFID will provide the spatial resolution for this. If the video quality is good enough, it would be doable to implement multi-animal Deep Lab Cut or SLEAP (www.sleap.ai) for multi-animal pose tracking. SLEAP in particular is very user friendly.

2. I thought that categorizing behaviors as "influencing" vs "following" events was very interesting and was curious if individual mice are influenced by particular other mice more than others? i.e. maybe animal 1 is highly influenced by animal 2, but animal 3 is more influenced by animal 4? Or is there one mouse that clearly influences the other 3 in a more generalized manner? Either way regardless of the outcome it would be very interesting to show the data.

3. The data set is quite a rich one due to the sheer length of time that the behavior was recorded for. It would greatly enrich the manuscript to show how preferences evolve (or remain stable?) across hours and different phases of the light-dark cycle. In the manuscript the authors wrote that they analyzed this, but do not seem to state the results or show the data. Unless I am missing something?

4. Did the authors rank the mice? If so, is there any correlation between social rank and preference for objects? Or any correlation between social rank and percent time influencing vs following? (If they did not rank the mice, I am not suggesting that they re-do the experiments with ranking).

Reviewer #3: Hobbiesiefken et al. address the critical question of what types of housing enrichment socially-housed laboratory mice engage with most. They name specific items with the highest worth values, but find substantial variation in preference among individuals and groups tested. This work has important implications for how mouse researchers and animal technicians care for their animals, which the authors give strong recommendations for in closing their discussion.

The statistical methods used appear sound given the researchers’ choices, but I see several alternate choices that would provide new opportunities for analyses that would round out the story or make the findings more behaviorally relevant. These recommendations are included in several comments below.

I appreciate the authors’ assertion that elements with low worth values or high consensus error may still be enriching for mice (line 251). An additional consideration is whether there was an aversion to any item. Does a low worth value indicate an aversion? What would be a threshold or difference to determine an aversion for any particular elements? Are statistical comparisons possible with the worth value data?

One consideration that the authors have nearly overlooked is the different dynamics of socially-housed mice compared to isolated mice. While social housing is and should be the norm, there are many conditions in which mice are housed individually (i.e. for research or medical purposes). Enrichment is arguably even more important for individually-housed mice, and preferences may be different under those circumstances. This should be addressed in the manuscript beyond explaining that singly housed animals were not used; I believe this warrants separate experiments, but at the very least this should be acknowledged in the discussion.

I have several issues with the analysis of influence within a cage.

First, there is an assumption that social influence would only positively influence the preferences of enrichment items – only “follows” are assessed. It is likely that there could be an effect in the opposite direction: one mouse interacting with an item could lead to another mouse choosing a different item – for example, due to avoidance of the cagemate, or to some logistical ease of playing with a separate toy.

Second, and related to the first, just looking at “follow” events is very limiting. A more relevant metric both to address influence and how mice engage with their items would be how much time they spend with at least one other mouse using the item at the same time.

Third, both the 1 and 3 second follow criteria are arbitrary and don’t have any rationale. Imagining a “follow” in under 1 second, I see one mouse blindly following another, practically in lock-step, regardless of what it was doing, which doesn’t seem natural. If instead I imagine a scenario where a “follow” seems natural, I imagine Mouse 1 moving to a new cage/enrichment object while Mouse 2 is doing its own thing, then notices 1 is gone and/or playing with a new object, then decides to follow, then stops its own activity, then follows. This could take many more than 3 seconds and still be socially relevant. See second point above for one alternative analysis, but there are others including expanding the time for analysis that I think would be more relevant.

I don’t understand how the statement “This supports the hypothesis that nest boxes are also perceived as important exploration objects for mice rather than a mere refuge and sleeping area” (line 298-300) follows from the preceding finding that mice had no real preference of housing enrichment during their inactive phase. It seems this shows instead that regardless of how a housing element is used during the active period, they will all be similarly used during the inactive period.

Line 315 – what are consumer demand tests?

A final important consideration for advising a huge change in housing conditions is how that may affect the mice (e.g. their brains) and how that could influence future research conducted with “enriched” mice, especially for comparison with past research.

Figs 1-2: What is the significance of the line connecting the items on the graph? The size of the dots? I think it would be helpful to see all the graphs have the same Y-axis range, and to have all of the item names be arranged in a single column rather than scattered, which makes the graph look like it is more dimensional than it is.

6. PLOS authors have the option to publish the peer review history of their article (what does this mean?). If published, this will include your full peer review and any attached files.

Reviewer #1: No

Reviewer #2: No

Reviewer #3: No

---

## [Author Response · Author response to Decision Letter 0]

9 Nov 2022

see attached response to reviewer document

---

## [Editor Report · Decision Letter 1]

22 Nov 2022

Rating enrichment items by female group-housed laboratory mice in multiple binary choice tests using an RFID-based tracking system

PONE-D-22-23528R1

Dear Dr. Lewejohann,

We’re pleased to inform you that your manuscript has been judged scientifically suitable for publication and will be formally accepted for publication once it meets all outstanding technical requirements.

Kind regards,

Andrey E Ryabinin, Ph.D.

Academic Editor

PLOS ONE
---

## [Editor Report · Acceptance letter]

24 Nov 2022

PONE-D-22-23528R1 

Rating enrichment items by female group-housed laboratory mice in multiple binary choice tests using an RFID-based tracking system 

Dear Dr. Lewejohann:

I'm pleased to inform you that your manuscript has been deemed suitable for publication in PLOS ONE. Congratulations! Your manuscript is now with our production department. 

Kind regards, 

on behalf of

Dr. Andrey E Ryabinin 

Academic Editor

PLOS ONE